# Machine Learning in Healthcare Communication

**Sarkar Siddique [1] and James C. L. Chow [2,3,*]** 

1. Department of Physics, Ryerson University, Toronto, ON M5B 2K3, Canada; sarkar.siddique@ryerson.ca
2. Princess Margaret Cancer Centre, Radiation Medicine Program, Department of Medical Physics, University Health Network, Toronto, ON M5G 1X6, Canada
3. Department of Radiation Oncology, University of Toronto, Toronto, ON M5T 1P5, Canada
* Correspondence: james.chow@rmp.uhn.ca; Tel.: +1-416-946-4501

**Definition:** Machine learning (ML) is a study of computer algorithms for automation through experience. ML is a subset of artificial intelligence (AI) that develops computer systems, which are able to perform tasks generally having need of human intelligence. While healthcare communication is important in order to tactfully translate and disseminate information to support and educate patients and public, ML is proven applicable in healthcare with the ability for complex dialogue management and conversational flexibility. In this topical review, we will highlight how the application of ML/AI in healthcare communication is able to benefit humans. This includes chatbots for the COVID-19 health education, cancer therapy, and medical imaging.

**Keywords:** artificial intelligence; machine learning; healthcare communication; chatbot

## 1. Introduction

Artificial intelligence (AI) is a computer program's capability to perform a specific task or reasoning processes that we generally associate with intelligence in human beings. Primarily, it has to do with making the right decision with vagueness, uncertainty, or large data. A large quantity of data in the healthcare field, from clinical symptoms to imaging features, requires machine learning algorithms for classification [1]. Machine learning is a technique that utilizes pattern recognition. AI has been implemented in several applications in the clinical field, such as diagnostics, therapeutic and population health management. AI has a considerable impact on cell immunotherapy, cell biology and biomarker discovery, regenerative medicine and tissue engineering, and radiology. Machine learning in healthcare applications are drug detection and analysis; disease diagnosis; smart health records; remote health monitoring; assistive technologies; medical imaging diagnosis; crowdsourced data collection and outbreak prediction; and clinical trial and research [2]. A large quantity of data, also known as big data, is now available to train algorithms [3]. Several algorithms consisting of Convolution Neural Network (CNN) of more than a 100 layers have been used to diagnose pneumonia conditions. Several studies show that several algorithms can perform at the same level as a clinician and in some cases outperform clinicians. Specialists are still needed, however, as they can ensure safety and monitor AI output. AI does not hope to replace clinicians but to assist them and make their job more efficient. Facial analysis technologies have the capability to perform at the same level as clinicians with the help of deep learning. The Food and Drug Administration (FDA) has granted approval for a significant number of proprietary algorithms intended to be used for image analysis and interpretation; a prominent example would be Aidoc used in radiology to detect intracranial hemorrhage [4]. AI is used heavily in medical imaging to help in rendering medical diagnoses. Machine learning in medical imaging typically starts with algorithms looking for image features it believes to be important and will yield better predictions. A decision tree is an algorithm system, it identifies the best combination of features to classify the image or compute specific metrics for

the image region. Several methods are also used for this purpose where each has its own weaknesses or strengths [5]. AI needs to be evaluated for stability and safety before it can be implemented in clinical settings. There have been many research papers on evaluating AI's decision-making and clinical decision support. The recent advances in AI and the introduction of PyTorch, DeepLearning4J, TensorFlow, and Keras have led to the development of numerous algorithms that is available to clinicians to implement in many applications [6]. Machine learning allows us to make informed clinical decisions through insights from past data and is the core of evidence-based medicine. AI provides techniques to analyze and reveal complex associations that are difficult to convert into an equation [7]. A neural network is a model aspiring to mimic how the human brain works. It is composed of large numbers of interconnected neurons. Machine learning can utilize this to solve complex problems by analyzing evidence to provide an appropriate conclusion. Machine learning can simultaneously observe and process at a very fast pace with almost limitless inputs. It carries a transformation pattern to healthcare supported by clinical data's extensive availability and recent advancement in analytics systems. Machine learning technique mimics medical practitioners in complicated problem solving through cautiously considering proof to make valid decisions [8]. AI and deep learning have been enabled by labelled big data and improvements in cloud storage modalities, and enhancement in computing power. In medicine, it has a significant impact at three levels: for the health system by improving workflow and the potential for reducing medical errors; for clinicians primarily through rapid, accurate image interpretation; and for patients by having the ability to process data, which ultimately promotes health [9]. Chatbots have high potential in a clinical setting, but for them to be used safely in clinical settings, they first need to be evaluated like a novel medical device or like a new drug [10]. Two very important factors in terms of patient care for physicians are knowledge and experience; however, in terms of gaining knowledge by cumulating data, humans are limited, but machine learning can excel in that area [11]. Machine learning can generally be classified into two categories: supervised learning and unsupervised learning [12].

## 2. Machine Learning Technology

### 2.1. Natural Language Processing (NLP)

Machine learning is computer-based, and its primary objective is to analyze free form text or speech that follows a predefined set of theories and technologies such as linguistic and statistical methods, which obtains rules and patterns from the analyzed data. It can convert the text into a structured format that follows a hierarchy. The itemized elements have a fixed organization and standardized terminology for each element. These texts are easily queried and manipulated. Pattern similarity and linguistic analyses are the primary NLP technologies used. Pattern matching is a straightforward and fundamental text searching technique. It is essential to form complex NLP. Stemming is always used in NLP that uses language morphology knowledge to convert a given word down to its simplest form or root. Stemming is, therefore, suitable for agglutinative languages, while other languages need lemmatization. Breaking texts into tokens or chucks is another application of pattern matching, also known as tokenizing. Linguistic NLP systems read words as a symbol combined based on grammatical rules, and usually rely on the assumption that words forming sentences/expression/texts are conceptual and meaningful. A computer algorithm uses both syntactic and semantic knowledge to infer what concept modifies another concept. With the era of deep neural networks, word n-gram features are replaced with the word or sentence embedding; however, similarity obtained from linguistics NLP as a feature in classification algorithm often gives better results [13]. Natural language systems are used in many important clinical and research tasks such as NLP-based computational phenotyping, like clinical trial screening, pharmacogenomics, diagnosis categorization, novel phenotype discovery, the interaction between drugs, and detection of adverse drug events [14]. Several natural language processing systems that are developed for clinical use are shown in Table 1 [15]:

**Table 1.** Natural language processing (NLP) application system [15].

| System Name | Brief Description |
| --- | --- |
| ASLForm | It is an adaptive learning system that has some fundamental rules for finding a target text. As a user selects output, it continuously and simultaneously updates. |
| COAT | It is a clinical note processing system that is rule-based and uses machine learning (through WEKA) components with the integration of MetaMap Transfer. |
| LEXIMER | It was implemented to render medical imaging and has the ability to find significant recommendations and clinical findings from CT and MRI reports. |
| Barrett et al. (unnamed) | It can identify 17 serious sentinel events such as sepsis, dyspnea, and delirium in palliative carte consult letters. |
| Martinez et al. (unnamed) | It takes NegEx, Genia Tagger, and MetaMap as input and can classify cancer staging pathology reports. |
| Otal et al. 2013 (unnamed) | It can detect T cancer staging classification. It uses WEKA. |
| Wieneke et al. 2015 (unnamed) | It can extract results, laterality and procedure from breast pathology reports, and if high NPV and high PPV classifiers do not agree, then it is sent for manual review. |

NLP is used to translate or map phrases or words onto concepts by tokenizing, lemmatizing, and mapping each lemma. By taking each token as a weighted linear combination of topics, like a generalized linear model as a dependent variable, which is a weighted linear combination of independent variables. It is called Latent Dirichlet Allocation, also known as topic modeling. Application of NLP in Free Open Access Medical Education is an excellent source of educational materials. Free Open Access Medical Education's resource includes websites, blog posts, or podcasts. Those interested in medical care can comment, discuss and provide insight to content relative to medical care. NLP can help organize Free Open Access Medical Education by identifying which topics are most prevalent. An algorithm can automatically rate each website, enhances Academic Life in Emergency Medicine, which relies on a panel of experts who review blog posts individually [16]. Gene name normalization involves list-wise learning, conditional random fields, graph-based normalization, regression-based methods, and semantic similarity techniques. The different application uses different methods. Some examples are using naïve Bayes machine learning system to link SNIMED to keywords or phrases from clinical records; Kate normalized clinical phrases to unified medical language system (UMLS) IDs using the pattern distance calculation method. Other applications use MetaMap, Knowledge Extraction System (cTAKES), or clinical text analysis to extract and map a term to concepts in UMLS. In 2015, the Clinical TempEval challenge was to extract temporal information from the clinical note. All of the systems used supervised classifiers to solve the challenge, and all of them outperformed the rule-based systems. Marine Health information exchange EMR database's clinical notes were used to train and test an NLP system. The system can find congestive heart failure cases with an F-score of 0.753. Machine learning was used to analyze electronic health records to evaluate the risk of suicide dependent on physical illness. Medicare providers and Centers for Medicaid have tried to evaluate patients that are at high risk for readmissions. These patients are followed up to receive appropriate intervention [17]. Shared task organized for the advancement of NLP in clinical data was formed to provide a good source of resources for future training purposes. It allows the global research community to take on challenges that would otherwise be out of reach. Clinical NLP shared task utilized social media data such as forum posts; Twitter, journal articles such as PubMed, and electronic health records such as psychiatric evaluation records, pathology reports, and nursing admission notes; and other health-related documents such as drug labels [18]. Clinical notes in neurology or in the field of cardiology

are useful for deriving metadata in the development of downstream machine learning applications. A pipeline was constructed using the clinical NLP system, UMLS, cTAKES, Metathesaurus, Semantic Network, and learning algorithms. The pipeline was used to extract features from two datasets. The study shows that a supervised learning based NLP approach is beneficial for developing medical subdomain classifiers [19]. A new prototype NLPPReViz was designed for clinical researchers to review, train, and revise NLP models. The prototype study involved nine physicians. The study group built and reviewed models with two colonoscopy quality variables. Using the initial training set as small as ten documents led to a final F1 scores variable between 0.78 and 0.91 [20]. The medical literature relevant to germline genetics is increasing at a rapid pace. As a tool that helps monitor and prioritizes literature to understand pathogenic genetic variants, their clinical implications are very beneficial. A study developed and evaluated two machine learning models to classify abstracts that are correlated to cancer proliferation for germline mutation carriers or the normality of germline genetic mutations. The first model utilizes a support vector machine (SVM) that learns a decision rule that is linear based on the bag of n-gram representation of each abstract and title. The second model is a CNN that learns a set of parameters based on the raw abstract and title [21]. Bao's study proposed a clinical text classification model. The model utilizes knowledge guided deep learning. The trigger phrases or words were found using rules, and they were used to label an example set. The example set was then later used to train a CNN. The result showed the CNN models effectively capture the knowledge of the domain and can learn hidden features by utilizing the target words and concept unique identifier (CUI) embeddings. Their model was able to outperform the state of the art methods in the i2b2 obesity challenge [22]. In maternal death, suicide can be prevented if quick appropriate action is taken to mitigate risk. A study was done to use NLP in electronic medical records to evaluate and find pregnant women with suicidal behavior. Three algorithms were validated to classify the suicidal behavior and obtain-curated features [23]. NLP in other languages is challenging because it is not as extensively explored as it in the English language; however, it offers many beneficial advantages such as text mining from the health record of non-English speaking countries. It also allows the chatbot to communicate with participants that do not speak English. Some of the languages where NLP has been explored are French, German, Spanish, Japanese, Chinese, Dutch, Swedish, Portuguese, and others. Jacob used deep learning to detect health-related infection in Swedish patient records. Lopprich described a system using NLP for the German language to classify multiple myeloma patients, and the study was conducted at Heidelberg University Hospital. Metzger showed that the development of machine learning-based classifiers that make use of free-text data could identify suicide attempts. The study was done in a French Emergency department, and the study result shows promising results [24]. A variety of NLP methods, with an overall shift towards deep learning, were observed in the last four years. Deep learning can automatically find mathematically and computationally convenient phrases or abstracts from raw data that can be used for classification without having explicit defined feature [25].

*2.2. Deep Neural Network (DNN)*

The computational power of deep learning has revolutionized modern AI. It led to the start of many new research companies in recent years [26]. Deep learning can automatically determine the parameters that are deep in a network-based, which is based on experience. Neural networks have multiple hidden layers that have been discovered for a long time but never have been explored to this degree before. In the 1980s, it was a popular idea in the field of cognitive science and frequently used in the field of engineering [27]. DNN can operate with supervision or no supervision. In machine learning, data complexity is often reduced to highlight the correlated patterns. DNN can achieve that, and it has the ability to learn input data order representation independently and generally requires a high volume [28]. Deep learning is artificial neural networks (ANN) having one or more hidden layers. It resembles the human brain due to how it functions. Deep learning can

achieve higher accuracy if trained on big data, especially in the medical field. Useful and important information from big data can be extracted with deep learning. ANN is the fundamental base of DNN. ANN is a classification and a regression algorithm and quite popular. It has multiple layers and computes by mimicking the transmission signal and has a human brain like neuron architecture and synapses. ANN is composed of artificial neurons that are interconnected. Each neuron uses activation functions to output a decision signal according to the weighted sum. In ANN, many computing units are merged. DNN is composed of stacked layers assembled in a series. The first layer is for input, while the last layer is for output, with hidden layers from the second layer to as many layers as required. DNN extends the depth of layers and yields improved prediction and performance with more complex layers in recognition studies [29]. Deep learning is used heavily for gene expression and classification. Forest DNN is a new classifier that was introduced, which integrates the DNN architecture with a supervised forest detector. It can overcome the overfitting problem by learning sparse feature representations and uploading the data into a neural network [30]. DNN is used in the prediction of protein interaction using primary sequences [31]. Deep learning is used in many areas of radiology. CheXNeXt based on CNN was developed to simultaneously detect 14 different pathologies, including pleural effusion, pneumonia, pulmonary masses, and nodules in frontal-view chest radiographs [32]. A DeepNAT, which is a 3D Deep CNN, was introduced to be used in T1 weighted magnetic resonance images to auto segment neuroanatomy. It learns abstraction feature and multiclass classification in brain segmentation by utilizing an end-to-end approach [33]. The structured report enhances communication between radiologists and providers. In order to convert unstructured computed tomography (CT) pulmonary angiography reports into a structured format, a deep learning algorithm was employed. The algorithm converted free text conclusion into structured reports efficiently and yielded high accuracy. Overall, it increases communication between radiologists and clinicians without any common major downfall such as losing productivity. It provides enhanced structured data for research and data mining applications [34]. Public health monitoring and management is an important aspect of government administration. It provides insights into the best strategies to implement. Health status monitoring and investigation has traditionally been conducted through surveys and responses from targeted participants; however, it is very costly and time consuming. Social media based public generated content related to health can be used to predict health outcome at a population level. It makes use of linguistic features to analyze text data. It also makes use of visual features by utilizing deep neural networks. Two large-scale online social networks were used for data collection: Foursquare and Flicker, in order to predict U.S. health indices. The experimental result concluded the prediction made from social data yields comparable results and outperform textual information [35]. For a majority of the people, speech is the primary method of communication and the use of a microphone sensor allows computer–human interaction. Using speech signals to quantify emotions is an emerging area of research in human–computer interaction. It can be applied to multiple applications such as virtual reality, healthcare, human reboot interaction, behaviour assessment, and emergency call centers, where being able to determine the speaker's emotional state from just speech is very beneficial. A system that is able to learn discriminative and salient features from spectrogram of speech signals using deep convolutional network was developed. Local hidden patterns are learned in convolutional layers that have special strides to down sample the feature maps rather than pooling layers, and full-connected layers are used to learn discriminative features. A SoftMax classifier was used to classify emotions in speech [36]. A bidirectional recurrent neural network containing the attention layer was used to develop a chatbot with Tensorflow, capable of taking a sentence input with many tokens and replicate it with a more appropriate conversation [37].

### 3. Application of Machine Learning in Healthcare Communication

*3.1. Overview of Chatbot*

A chatbot is a form of a computer system that lets humans use natural human language to interact with computers. Some examples of modern chatbot systems are AliMe, DeepProbe, SuperAgent, MILABOT, and RubyStar [38]. ELIZA is the first chatbot developed by Professor Joseph Weizenbaum at the Massachusetts Institute of Technology. ELIZA showed that communication with a computer is possible by utilizing language conversion. Key features of ELIZA were keyword spotting and pattern matching with 200 stimulus-response pairs. A.L.I.C.E. was developed as a modern version of ELIZA that first surfaced in 1995. A.L.I.C.E. had quite a lot of interesting features, such as case-based reasoning that extracts the correct context of ambiguous words, knowledge base both temporal and spatial, random sentence generator, spell checker, and 45,000 stimulus-response pairs. VPbot was later introduced as a SQL based chatbot to be used for medical applications. VPbot uses a relational data model to store "language rules". VPbot focused on a targeted topic of conversation. The VPbot algorithm takes three input parameters, a vpid (unique identifier of each VPbot instance), the current topic, and a sentence. The VPbot exports a new sentence and a new topic [39]. A chatbot can recognize the user input through many forms and access information to provide a predefined acknowledgment [40].

*3.2. Patient Care*

Healthcare chatbots have high potential in medical communication by improving communication between clinic–patient and doctor–patient. It can help fulfill the high demand for health services through remote testing, monitoring of medication follow-up, or telephone consultations. A chatbot can conduct fast and easy health surveys, set up personal health-related reminders, communicate with clinical teams, book appointments, and retrieve and analyze health data. Chatbots can provide fast or instant responses to patients' healthcare-related questions while looking for specific symptoms or patterns in predicting disease. An example would be the Internet-based Doc-Bot, which operates via mobile phone or Messenger. The bot can be tailored for specific health conditions, populations, or behaviors [41]. Chatbots' bidirectional exchange of information with patients can be used to screen treatment adherence or to collect data. They can be applied through several methods, such as text-based services like chat rooms, text messaging, mobile applications, or audio services, such as Alexa, Siri, Google Assistant, or Cortana. The iDecide chatbot can deliver information about prostate cancer, such as risk factors, epidemiology, treatment options, and side effects. A study was done to observe the effect of iDecide on prostate cancer knowledge on African American men aged 40 years and over with a prior history of prostate cancer. The study result showed significant improvement in prostate cancer knowledge after using the iDecide bot. A significant number of cancer patients are likely to face severe anxiety or depression. A web-based chatbot, Woebot, for cognitive-behavioral therapeutics was studied. The study's objective was to determine the acceptability, feasibility, and preliminary efficacy of a fully automated conversational agent to deliver a self-help program for people with anxiety and depression symptoms. The result showed the patients that engaged with Woebot significantly reduced suppressed depression during the study period, where the control group showed no change in depression levels. In radiation, an oncology chatbot can collect patient-reported outcomes during and after the treatment in a convenient manner. Algorithms can be used to red-flag the most at-risk patients or trigger an additional consultation with the physician and enroll in supportive care treatments [42]. To live a healthy life, accessibility to good healthcare is essential. Sometimes it is not easy to consult with a doctor due to availability and cost. A medial chatbot was proposed to provide a solution without having to consult a doctor. Chatbots can act as a medical reference book that can provide patients with further information about their disease and appropriate short-term preventive measures. For the chatbot to be viable, it is vital that the chatbot can diagnose all kinds of diseases and provide the necessary information. The bot interacts with patients and obtains their

medical issue through conversation and gives personalized diagnosis that corresponds to their symptoms. It will allow them to have the right protection [43]. A chatbot was used as a medical consultant. It used doctorME application to gather treatment records and symptoms information [44]. Another chatbot design that was proposed can provide an answer for any query based on the FAQs using Latent Semantic Analysis and Artificial Intelligence Markup Language (AIML). The answers were primarily accurate and efficient. AIML was used to answer general questions like greetings, which follows a template, and service-based questions were answered using latent semantic analysis [45]. Mandy is a primary care chatbot that can automate the patient intake procedure and help healthcare staff. The chatbot carries out an interview, tries to understand the patient's complaints, and provides reports to the doctors or healthcare professional for further analysis. The system provides a mobile app to the patients, which is a diagnostic unit and a doctor's interface that can be used to access patient information. The diagnostic unit has three main parts: an analysis engine that takes the patient's symptom information and analyzes them, a symptom-to-cause mapper for reasoning about potential cause, and a question creator to produce further questions. The system is a knowledge-powered data-driven natural language processing modality [46]. Chatbots function similarly to how a search engine works; however, the chatbot provides one answer where search engine provides multiple outputs. The chatbot is focused on becoming a search engine for the next task, which correlates to the previous task, which makes its input processing efficient. What makes a chatbot efficient and gives the feeling of an actual human is its extension and prerequisite enabled relations between responses. A chatbot can provide various responses to match the chat context [47]. Traditional chatbots lack the capability to have conversations with respect to the social context. Dialog considers both social and individual processes. Augello proposed a chatbot model that is able to choose the most suitable dialogue plans according to appropriate social practice [48]. The research in human–computer interaction on the chatbot dialogue system equipped with an audio-video interface shows promising results. They are equipped with catchy interfaces using human-like avatars capable of adapting their behavior to match the conversation context. They can vocally communicate with users through a text to speech systems and automatic speech recognition. The visual aspect of interaction gives it the ability to synchronize speech with an animated face model. It plays an important role in human–computer interaction. These kinds of systems are called talking heads [49]. A chatbot often acts as a virtual assistant. It can have its virtualization. Its conversational skills and other behavior are simulated through AI [50]. Chatbot technology generally uses pattern matching to analyze the user's input. It uses templates to find and release output. Conversational interfaces these days are built to communicate like a human and try to mimic human conversation as much as possible. The automated online assistant is designed to assist, or even replace humans in some cases, and provide the service on its own [51]. The ability to connect with patients with a proper understanding of their situation is very important in AI development. There is a study regarding breast cancer patients communicate and respond to chatbot Vik. Vik communicates via text message and responds to any concerns or fears and gives prescription reminders. Human companionship and empathy is not replaceable but of the 4737 patients studied, 93.95% of the patients recommended Vik to their companions. This study shows that patients desire personalized focused support and care [52]. Kbot is a knowledge enabled personalized chatbot introduced for asthma self-management [53]. Nurse Chatbot can provide chronic disease self-management support. It can simulate anatomy (judgement) by super positioning. It has the ability to exhibit self-organization, cognitive and emotional response and makes communicative healthcare management possible. Some of its functions are setting goals for modifying behavior based on self-efficacy. It can offer options to make the patient adhere to medications and suggests healthy behaviors based on prediction. Laranjo's review of healthcare chatbots shows a significant effect of depression reduction due to the use of a chatbot. Patients that received chatbot care are expected to follow medication regimen better, healthy behaviors, and control

the symptoms better. According to Tanioka's review, humanoid robots can be a viable option to care if they can deeply empathize, observe, understand/judge, and be responsive to condition change, and have the ability to personalize care, normalize the discourses related to safety and ethical problems. The latest chatbots running on neural network can achieve real human response due to their dialogue being a hybrid of rule-based and generation-based model. AI in nurse chatbot is inspired by Froes's view of biological cognition. It had a pre-reflective process acting on the training data sets similar to text corpus, feedbacks, and chat. Deep learning function is used to extract patterns and their occurrence was predicted using a recurrent neural network algorithm. Reflective, on the other hand, is used in NLP, where text inputs were lemmatized by removing prefixes without changing the core meaning [54]. PARRY was another chatbot that was developed, which can simulate a paranoid patient [55]. Interest in mental health chatbots has gained considerable attention where many companies labeled them "the future of therapy" [56]. A digital chatbot to support nurses and prevent risk of hospitalization for older adults was also explored and the study shows promising result [57].

*3.3. Radiology and Radiotherapy*

An AI-based online support group was proposed that uses machine learning NLP to automatically cluster patient data. Patient behaviors, demographics, decisions, emotions, clinical factors, and social interaction were extracted and analyzed. Patient-centered information about all medical illnesses and conditions are resourced into online support groups. A recent study shows that a majority of the people, approximately more than 80% of the user with Internet access, look for medical-related information through online medical resources or social media. Online support groups provide comfortable virtual spaces with animosity for patients, careers, and general information seekers. It allows them to obtain advice, express emotions, and share experiences. Online support group discussions are discussion threads. The thread starts with a question, experience, or comment related to a patient's health issue. In online support group posts, patients often mention their clinical information, relevant decision factors, demographics, decision-making process, and emotion. The timeframe of emotional and clinical information is stored in the post's timestamp and generally mentioned in the post content. This information in social media posts is often hidden in a large amount of unstructured text data. Advanced machine learning, deep learning, and NLP can offer solutions to this problem. A Patient-Reported Information Multidimensional Exploration framework was proposed to automatically analyze patient behaviors, patient emotions related to diagnosis, clinical factors, and treatment and recovery [58]. Radiology educational post on social media is beneficial. A study was conducted with 2,100 responders from 124 countries, with most of them aged below 40. The study result shows that the participants have found that the radiology education posts on social media was useful [59]. An oncology-based chatbot, OntBot, was developed. OntBot uses suitable mapping techniques to transform knowledge and ontologies into a rational database and then use the database to run the chat. This approach overcomes several traditional chatbot hurdles, such as the use of chatbot specific language (AIML), high bot master interference, and premature technology. OntBot has an easy interface that makes use of natural language and great support for the user [60]. Cloud computing is a set of technologies that offers storage and computing service. This highly popular computing system can significantly enhance dose calculation efficiency using Monte Carlo simulation in radiation treatment planning that involves complex and intensive mathematical computation [61]. A graphical user interface using Monte Carlo simulation for the external beam treatment planning was studied and showed brilliant machine learning implementation. The Monte Carlo simulation was carried out in parallel using multiple nodes in a high-performance computing cluster [62]. Different imaging modalities, such as computed tomography (CT), magnetic resonance imaging (MRI), single-photon emission computed tomography (SPECT), and positron emission tomography (PET) data integration will yield more accurate and detailed images. It will be very beneficial to have the modali-

ties automatically recognize the patient's organ in the treatment using various machine learning techniques, such as pattern recognition. Such a smart machine learning algorithm can be developed using minded data. Machine learning methods can take advantage of the big data cloud. Deep learning allows large-scale image recognition tasks, and it is done through a series of pooling, classification, and convolutions. In treatment planning, a knowledge-light adaptation idea was explored using case-based reasoning system. The idea was tested on 3D conformal treatment planning for breast cancer patients. The study result showed the neural networks based system increased the success rate by 12% while the adaptation-guided retrieval of the case for beam number improved the success rate of the system by 29% [63]. An app for smartphones or tablets to calculate the monitoring unit in superficial and orthovoltage skin therapy was developed with the frontend window as shown in Figure 1. The app can run on both Android and Windows operating systems and can calculate the monitor unit. It takes advantage of all-natural features of the Internet, such as live communication, access to the Internet, data sharing, and transfer. The app loads the available mean energies. The app uploads and displays the available cones or applicators compatible with the selected energy. The user needs to input the desired cone from the availability list, prescribed dose, and number of fractions the dose will be delivered in. The app will find and provide the treatment faction dose or daily dose [64]. Machine learning can predict dose-volume parameters, such as the dose distribution index in radiation treatment planning quality assurance. In order to determine the parameter efficiently and effectively, the machine learning algorithm needs to be selected appropriately with performance, as shown in Table 2 [65]. Telehealth or telemedicine is health resources or services administered through telecommunications or electronic information technologies. It supports long-distance professional and patient health-related education and clinical health care. Telehealth allows remote bidirectional information flow and visual and audio interaction with patients. Telephone follow-up can be used to address the psychological aspect of disease and treatment. Studies in various cancer subgroups are acceptable to have telephone follow-up in prostate, brain, endometrial, colorectal, and bladder cancer. Potential cost benefits studies also studied the impact of telehealth and how it can help to manage cost in healthcare and overall improve quality of life for patients. One study has examined remote symptom monitoring in 405 cancer patients with depression or cancer-related pain or both. With the use of a telehealth system, patients with depression had greater improvements combating depression and patients with cancer-related pain had impactful brief pain relief compared to control [66]. Vivibot is another chatbot developed to deliver positive psychology skills and influence a healthy lifestyle in young people after cancer treatment [67]. It is another chatbot that was introduced to promote physical activity and a healthy diet was [68].

### 3.4. Education and Knowledge Transfer System

In order to provide a practical collaborative healthcare workforce, interprofessional education is essential. Interprofessional education occurs when more than one profession learns with, or from, or about each other. Its main purpose is to increase collaboration and improve healthcare quality. It provides a collaborative framework and provides insight into how each discipline contributes without losing its own identity [69]. Text mining and computational linguistics are two computationally intensive fields where more options are becoming available to study large text corpora and implement corpora for various purposes. In these systems, the analysis of text does not require a deep understanding technique. AIML is the assembly language for the AI conversational agent, such as a chatbot in most cases. A study uses corpora seed to extract relevant keywords, glossary building, text patterns, and multiword expressions to build AIML knowledge and use it to build an interactive conversational system [70]. High-quality information can be extracted from an online discussion forum that can be used to construct a chatbot. In one of the studies, the replies that were logically related to the thread title of the root message are obtained with a SVM classifier from all the replies based on correlation. Then the extracted

pairs were ranked based on content qualities. Finally, the top N pairs were obtained to be used as chatbot knowledge [71]. In the last few years, there was a rapid increase in chatbots in various fields, such as health care, education, marketing, cultural heritage, supporting system, entertainment, and many others. The chatbot in the educational domain has high potential, especially when delivering medical science knowledge. Fabio's paper shows the implementation of such a chatbot as a teaching medium and demonstrated its utility [72]. A chatbot can be used in instructional situations for educational purposes due to its interactive ability as opposed to traditional e-learning modules. Students can continuously interact with the chatbot. They can ask questions related to a specific field. A chatbot can also be used to learn or study a new language. It can be used to visualize a corpus's content, a tool to access information systems, and a tool to give answers to questions in a specific domain. Some chatbots can be trained in any language or any text [73]. Chatbots have high potential in distance education. A chatbot named Freudbot was introduced. It was constructed with the open-source AIML, and its main purpose was to increase student interaction in distant education. Fifty-three students in a psychology class completed a study where they chatted with Freudbot for 10 min via the web. They then provided feedback about their experience with the chatbot and demographic description. The questionnaire result shows a neutral evaluation of the chat experience. However, the participants positively encouraged the further exploration of chatbot technology. They also provided clear feedback for efficient future development and improvement. The chatbot shows high potential in distance education [74]. Chatbots are also used in other domains, like a dialogue-based natural language chatbot that can act as an undergraduate advisor [75]. As a computer simulation in a scholarly communication system, its primary goal is to provide a virtual chatting companion. It generated a communicative response corresponding to the user input. The dialogue context is generated from its personality knowledge, inference knowledge, and common sense knowledge [76]. A stress management chatbot was introduced to deliver a brief motivational interview in web-based text massaging format to relieve stress [77].

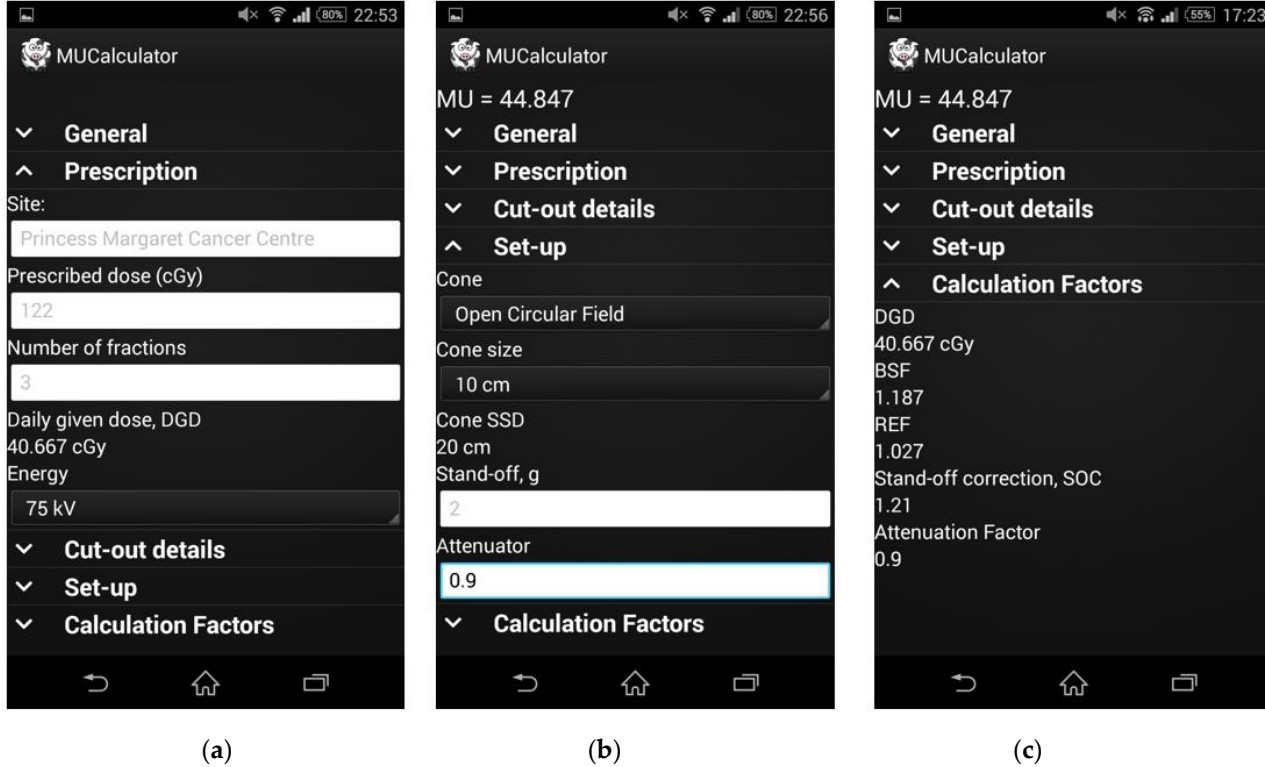

(**a**)  (**b**)  (**c**)

**Figure 1.** Android version of the app showing inputs of the (**a**) prescription, (**b**) beam setup, and (**c**) calculation factors and how the app automatically updates as information is input by user [64].

**Table 2.** Table summarizing the root mean square error (RMSE), R-squared, prediction speed, and training time of models created in the Regression Learning App available in the MATLAB's Machine Learning and Statistical Toolbox, using five dose-volume points from each dose-volume histogram (DVH) with 4-fold cross validation. They are ordered from the best performance to worst [65].

| Machine Learning Algorithm | RMSE | R-Squared | Prediction Speed (Observation/s) | Training Time (s) |
|---|---|---|---|---|
| Square Exponential GPR | 0.0038 | 0.99 | 4100 | 0.18 |
| Matern 5/2 GPR | 0.0038 | 0.99 | 3800 | 0.21 |
| Rational Quadratic GPR | 0.0038 | 0.99 | 2700 | 0.23 |
| Linear Regression | 0.0045 | 0.98 | 1700 | 0.37 |
| Exponential GPR | 0.0125 | 0.87 | 3900 | 0.18 |
| Linear SVM | 0.0123 | 0.87 | 4500 | 0.21 |
| Quadratic SVM | 0.0151 | 0.81 | 3400 | 0.13 |
| Cubic SVM | 0.0193 | 0.68 | 4700 | 0.11 |
| Fine Tree | 0.0218 | 0.60 | 4600 | 0.10 |
| Medium Tree | 0.0305 | 0.21 | 4600 | 0.42 |
| Coarse Tree | 0.0344 | 0.00 | 5600 | 0.09 |

AI has much potential, but it can also have profound health effects due to sample size and misclassification, underestimation, and measurement error [78]. If an AI is not safe, it should not be trusted. To assess trust, safety, interfaces, procedures, oversight, and system-level workflow, collaboration is needed between AI systems, administrators, patients, and clinicians. At the current state, conversational AI is not ready to completely replace human therapists anytime soon. Training the AI model for psychotherapy sessions is a difficult task. Psychotherapy sessions are rarely recorded, and if they are recorded, they pose a considerable risk to the patient. Not just in psychotherapy but also with any medical data, confidentiality is a big problem since a breach of confidentiality could seriously harm the patient. AI can reduce costs significantly and make the therapy sessions more engaging to people who experience stigmatization from talk therapy. It also allows easy access to healthcare in rural areas. In psychotherapy, clinician–patient relationship is significantly important. It is essential for the patient to feel comfortable engaging in a conversation with AI. The responses need to be very critical since inappropriate responses can significantly impact a patient's health. Suppose the patients do trust the AI and engage in a conversation—in that case, they are more likely to disclose sensitive information to AI, which they might not feel comfortable disclosing to a human or unresponsive clinician. In those situations, AI has the potential to outperform clinician [79]. A study was conducted to evaluate one year of conversation between breast cancer patients and a chatbot. The result shows it is possible to obtain support through a chatbot since the chatbot improved the medication adherence rate of the patient [80]. A supportive chatbot is also being studied in other health areas, such as a supportive chatbot for smoking control [81].

### 3.5. Emergency Response and COVID-19

In emergencies, a few minutes can save a life. Quicker health care and accessibility systems can save many lives in many cases. A semi-automated emergency paramedical end-to-end response system was proposed. It can distribute medical supplies on-site in case of emergencies. This system utilizes decentralized distribution and does not involve any third-party institutions to ensure security. The response system can be used in urban, semi-urban, and rural areas. It allows community hospitals to provide specialized healthcare despite the absence of a specialized doctor. K-nearest neighbor, SVM, and ANN are some of the classifiers that are used in the response system. The response system uses drones to access remote areas that are difficult for a human to access. The deep neural network allows the drone to detect objects, making it more accurate and reducing failure significantly. The chatbot takes user responses and evaluates them, then, if need be, passes them to the administration. The chatbot uses NLP to refine input responses. The response system can also be utilized to detect stroke using CT scans. The model is trained using previously identified data. The difference in the distribution of density and texture of tissue outlining the stroke region is used. The gray level Covariance matrix is the feature extraction algorithms that are used to extract the features [82]. With the introduction of surgical robots, remote telesurgery has been a strong motivator. The ability to deliver surgery over long-distance gives rise to more availability for surgery and improved surgical outcomes. A large number of procedures are performed every year with this technology. The majority of them are performed only over several meters; however, some surgeons have to transition to a long-distance infrastructure with this system. The daVinci system could be operated at distances as far as 4,000 km from the operating room [83]. HarborBot was a chatbot introduced to assist patients with social needs in a critical situation in the emergency department. Most emergency departments do not have extra staff to administer screeners, and response rates are low, especially for low health literacy patients. HarborBot can facilitate engagement with low health literacy patients and provide a solution [84].

Coronavirus disease 2019 (COVID-19) has become a major global concern since January 2020. It primarily affects the cardiovascular system and requires sensitive, fast, and specific tools to identify the disease early, and better preventative measures can be applied [85]. COVID-19 is a worldwide crisis. Over 100 million people have been infected already. It has caused over 2 million deaths worldwide. Many countries have overstretched their healthcare resources to mitigate the spread of the pandemic. AI was implemented to monitor and control the COVID-19 pandemic in several critical areas. It was also used to predict the risk of developing the disease, hospital admission, and progression in those areas. AI was also used for early detection and diagnosis. DNN can be used in conjunction with x-ray or CT to diagnose COVID-19 detection and automate the process to keep up with the overwhelming demand [86]. AI played a significant role in developing novel text and data mining techniques to aid COVID-19 research. It was used in CRISPR-based COVID-19 detection assay, the taxonomic classification of COVID-19 genomes, discovering potential drug candidates against COVID-19, and survival prediction of severe COVID-19 patients [87]. Machine learning can differentiate the critical group vs. the noncritical group just using patient data. It is essential for managing such a high demand for healthcare resources. Some of the algorithms used for analysis were random forest classification, ANN, classification and regression tree [88]. The rapid spread of COVID-19 means the government and health service providers had limited time to design and plan effective and efficient response policies. It is important to quickly obtain an accurate prediction of the vulnerability of geographic regions, such as countries that are vulnerable to this virus' spread. AI was used to make that prediction by developing a three-stage model using XGBoost, a machine learning algorithm that can estimate potential occurrences in unaffected countries and quantify the COVID-19 occurrence probability [89].

On 11 March 2020, the World Health Organization stated that the novel COVID-19 is a pandemic. COOPERA is a chatbot based healthcare system designed to be used for personalized smart prevention and care developed using the LINE app. It was used to

evaluate the epidemiological situation in Japan. The result shows a significant positive correlation between self-reported fevers and the reported frequency of COVID-19 cases. It suggests that a large scale monitoring of fever will help estimate the COVID-19 epidemic scale in real-time [90]. To control the spread of COVID-19, many governments implemented phone hotlines for potential case prescreening. These hotlines were overloaded with callers' volume, leading to a waiting time of several hours or even an inability to contact or receive health care at all. Symptoma is a digital health assistant program that can identify more than 20,000 diseases with high accuracy (greater than 90%). Symptoma was tested to identify COVID-19 using a set of clinical case reports of COVID-19. The study result showed that Symptoma could accurately differentiate COVID-19 in 96.32% of clinical cases. Symptoma allows free text input in 36 languages. Combined with the result and accessibility give Symptoma the potential technology to fight against COVID-19 [91]. Chatbots played a crucial role in combating COVID-19 (Figure 2). The chatbot takes the user request and identifies the massage pattern. Depending on the input response, AIML logic retrieves keywords related to the symptoms to analyze the existing medical conditions. The function of the chatbot can be split into two sections: request analysis and return response. First, the chatbot used predefined questions and feedback to analyze and evaluate the virus severity. If the user fails to define a precise answer, the bot will fail to provide the correct response; however, if the user does provide a valid and precise response, the bot returns a response of the patient's condition in the generic text. Render questions can often help precisely understand the user's request. The bot begins the conversation with an initiation question to engage the user in a natural conversation and identify preliminary COVID-19 symptoms from the user's location. Then, the bot displays whether the person is likely to be affected or not. After the initiation chat, the bot will ask for symptomatic information. After collecting the necessary information, the bot will find the severity percentage the user experienced and act accordingly either by contacting health specialists or providing information regarding immediate preventative measures. Most of the question is yes- or no-based question that can be easily implemented into a decision tree. The data can be stored from the web using dynamic XML [92]. Several web-based COVID-19 symptoms checkers and chatbots have been developed; however, some studies show that their conclusion varies significantly. A study was done to evaluate the performance of these symptom checkers. The result shows different symptoms checkers showing different strengths in terms of sensitivity and specificity. Only two checkers had a harmonic balance between specificity and sensitivity [93]. With the advancement in NLP, the chatbot can answer the user's questions automatically. However, these models are rarely evaluated and need to be fully explored before they can be implemented in a clinical setting. A paper proposed to apply a language model that can qualitatively evaluate the user's response and automatically answer COVID-19 related questions. They applied four different approaches: tf-idf, BERT, BioBERT, and USED to filter and retain relevant sentences in the responses. With the help of two medical experts, the BERT and BioBERT, on average, have been found to outperform the other two in sentence filtering tasks [94]. Deep learning is very beneficial in the fight against COVID-19; however, deep learning requires massive data, which is problematic for the recent COVID-19 case. A study proposed a system that boosts COVID-19 detection; it uses a data augmentation model that makes CNN more efficient by increasing its learning capability [95]. Machine learning was also used to examine COVID-19 related discussion, concerns, and sentiments using 4 million Twitter messages related to the COVID-19 pandemic [96]. A study was done to observe how people respond to COVID-19 screening chatbots. The study was done with 371 individuals that participated in a COVID-19 screening session and concluded that the primary factor that drove the user's decision was whether they trust the capability of the chatbot or not. Overall, the chatbot providing high-quality service is critical [97]. As the COVID-19 pandemic continues, the mental health of both the infected and noninfected is an important concern. A study administered survey data using a chatbot onLINE in Japan's most popular social networking service. It showed that people with COVID-19 patients

in close settings had higher psychological distress levels than those without. Prioritizing mental health and psychosocial support implementation tailored to close relatives, family, and friends of COVID-19 patients is very important [98].

Partners HealthCare (Mass General Brigham) implemented an automatic prehospital chatbot that directs the patient to an appropriate care setting before showing up at the emergency department and clinics. It saves resources, minimizes exposure to other patients and staff, and further exacerbates supply and demand mismatching. An AI chatbot was deployed through a voice response message that the COVID-19 hotline caller received while on hold. It instructed the caller to visit the Mass General Brigham website hosting the chatbot with the workflow as shown in Figure 3. Educational material following Centers for Disease Control and Prevention (CDC) guidelines was provided to the users with reassurance for the first category of callers. The second category of callers was sent to follow the instruction for further clinical evaluation. Being able to use a chatbot while on hold increased the efficiency of the nurse-hotline. The nurse and the caller are able to have an informed discussion, and it allows the nurse to operate efficiently and quickly to come to a mutually agreeable plan. The chatbot was made available on the Mass General Brigham website so patients could access it before calling the hotline and decrease the call volume [99].

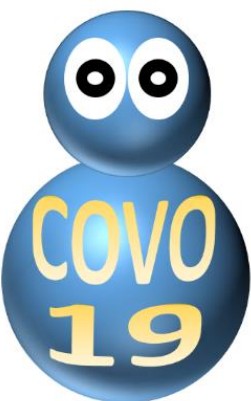

**Figure 2.** The COVID-19 Q&A Chatbot in Chow's research project.

Healthcare workers' screening for COVID-19 symptoms and exposure before every shift is essential for the spread of infection control. In order for this to work, the screening process needs to be efficient and straightforward. The University of California, San Francisco Health designed and implemented an AI-based chatbot-based workflow. It conducted over 270,000 screens in the first two months of use. It reduced the waiting times for employees entering the hospitals, yielded better physical distancing, prevented potentially infected individuals from coming to the hospital, and provided important live data for decision making staff [100].

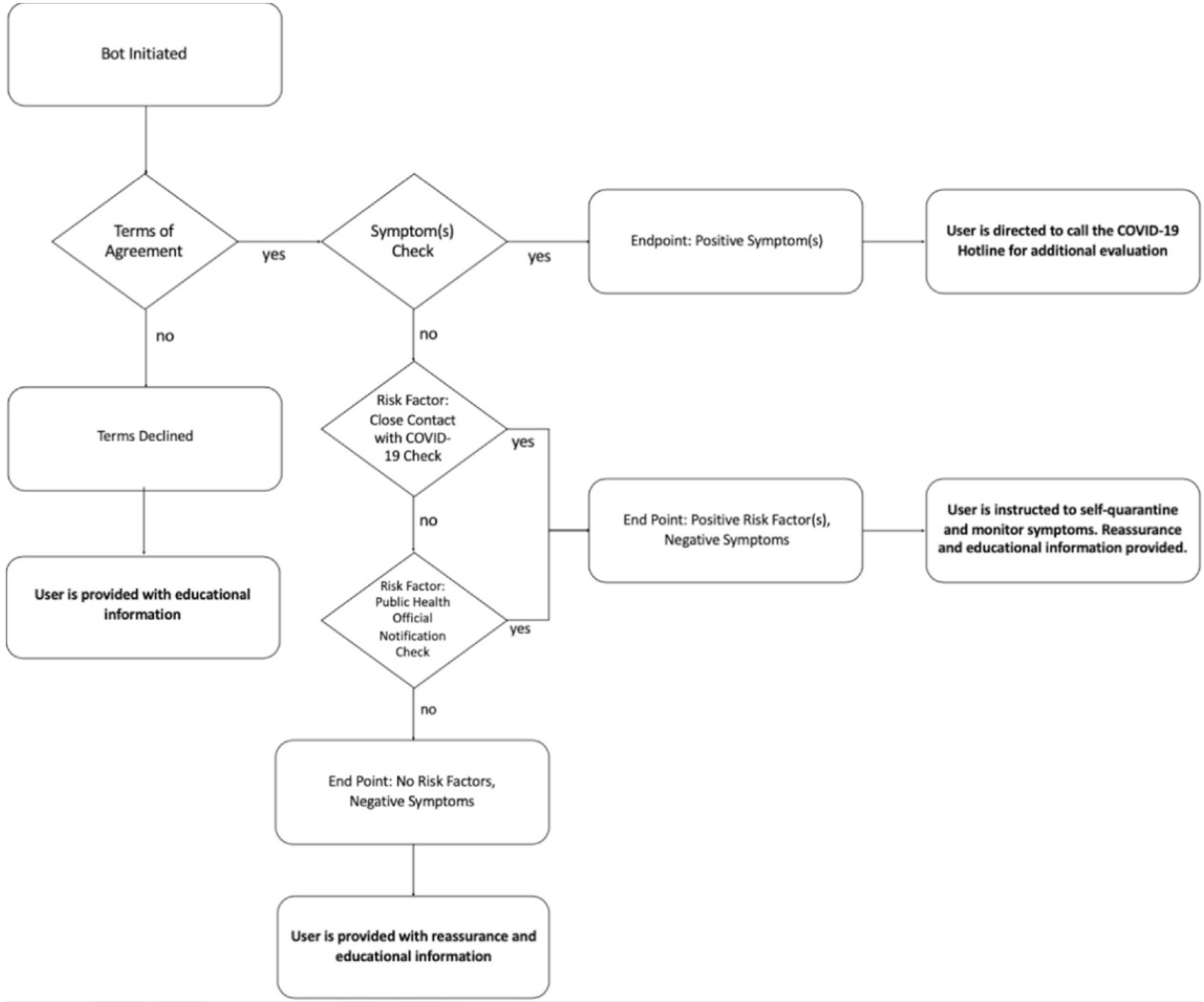

**Figure 3.** Flowchart of the Mass General Brigham chatbot system [99].

## 4. Conclusions

Machine learning is a robust and powerful digital tool that can benefit healthcare communication with a better patient care/education, faster decision making, and reduction of resource. To date, different fields in machine learning such as NLP and DNN have been studied and applied in various components of healthcare. The innovative AI-based chatbot takes an important role as a humanlike conversational agent between the user and service provider. This chatbot and response system has a significant impact on our healthcare system. With the continuous advancement in technology, it is expected to have more influence in the future. AI can reduce healthcare costs and make research tasks more efficient by introducing the latest advanced algorithm and is expected to assist clinical in many areas.

**Author Contributions:** Conceptualization, J.C.L.C.; methodology, J.C.L.C. and S.S.; resources, S.S.; writing—original draft preparation, S.S.; writing—review and editing, J.C.L.C. and S.S.; visualization, J.C.L.C. and S.S.; supervision, J.C.L.C.; project administration, J.C.L.C. All authors have read and agreed to the published version of the manuscript.

**Funding:** J.C.L.C. received financial support from the Planning and Dissemination Grants—Institute Community Support, Canadian Institute of Health Research, Canada.

**Acknowledgments:** J.C.L.C. would like to acknowledge the financial support from the Planning and Dissemination Grants—Institute Community Support, Canadian Institute of Health Research, Canada. J.C.L.C. would also like to thank the supports from Leslie Sanders and Kay Li from the York University, Toronto, Canada.

**Conflicts of Interest:** The authors declare no conflict of interest.

**Entry Link on the Encyclopedia Platform:** https://encyclopedia.pub/8078.

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
