# Peer review of "Machine Learning in Healthcare Communication"

_encyclopedia, doi:10.3390/encyclopedia1010021_

Round 1
Reviewer 1 Report
Machine Learning in Healthcare Communication
Figure 1: Flowchart of the Mass General Brigham chatbot system, the quality of the figure must be improved. If the information from the figure was taken from any sources, citation is required. The title of figure should be put below the figure.
The authors should use tables and figures to represent information instead of using a lot of text in this paper. That’s boring to read and not effective. Try to summarize information in tables and figures.
Besides, divide each section into several sub-section like 2.1, 2.2. 3.1, 3.2…..
Author Response
Authors’ Responses
Reviewer 1
We would like to thank the Reviewer for spending time on improving this work. Our answers to the comments are marked in blue fonts while corrections in the revised manuscript are marked in red fonts.
Machine Learning in Healthcare Communication
Figure 1: Flowchart of the Mass General Brigham chatbot system, the quality of the figure must be improved. If the information from the figure was taken from any sources, citation is required. The title of figure should be put below the figure.
Authors: The resolution of Figure 1 (Figure 3 in the revised manuscript) was increased. The source is from [99] which is on open access. The figure caption can now be found below the figure.
The authors should use tables and figures to represent information instead of using a lot of text in this paper. That’s boring to read and not effective. Try to summarize information in tables and figures.
Authors: We added two more figures (Figures 1 and 2) and one more table (Table 1) to the revision.
Besides, divide each section into several sub-section like 2.1, 2.2. 3.1, 3.2…..
Authors: Done.

Reviewer 2 Report
- The manuscript covers the problem quite well.
- The references are used very convincingly. However, it would be better if the authors analyzed the concise, concise sections 4,5,6.
- The images should use the best resolution.
- Part 8 needs to make more expansion.
Author Response
Reviewer 2
We would like to thank the Reviewer for spending time on improving this work. Our answers to the comments are marked in blue fonts while corrections in the revised manuscript are marked in red fonts.
The manuscript covers the problem quite well.
The references are used very convincingly. However, it would be better if the authors analyzed the concise, concise sections 4,5,6.
Authors: We have divided and numbered the sections and subsections as per the reviewer’s comments.
The images should use the best resolution.
Authors: We have increased the resolution of the figure in the revision.
Part 8 needs to make more expansion.
Authors: We expanded Part 8 (Conclusion Section) as follows:
“Machine learning is a robust and powerful digital tool that can benefit healthcare communication with a better patient care/education, faster decision making and reduction of resource. To date, different fields in machine learning such as NLP and DNN have been studied and applied in various components of healthcare. The innovative AI-based chatbot takes an important role as a humanlike conversational agent between the user and service provider …”
Reviewer 3 Report
The research paper is an overview of different ML tools used in Healthcare. Despite it contains important information on this topic, some concepts (about the deep learning methods) are not clearly presented. The paper needs to be restructured to be easily understandable.
My comments:
- Line 53 “A neural network works similar to how the human brain works”. Well, the artificial neural network is a very very simplified human brain model.
- Line 80. Stemming is suitable for EN or agglutinative languages; the other languages need lemmatization.
- Line 83. Linguistic NLP systems (especially word/sentence embeddings-based) usually rely on the assumption that words forming sentences/expressions/texts are conceptual and meaningful.
- Line 85. With the era of deep neural networks word n-gram features are replaced with the word or sentence embeddings.
- Line 139. CNN does not learn “non-linear complex decision rules”. This neural network approach (as any neural network method) learns only a set of parameters.
- Line 142. Any neural network (CNN, FFNN, RNN, etc.) is a typical supervised ML example that needs the annotated data. Despite if the preparation of the training data is manual or automatic (using rule-based approaches), it does not mean that CNN uses rule-based features. Please, clarify the explanation.
- Line 172 I cannot agree with the statement: “Deep learning is a section of machine learning that is composed of multiple artificial neural networks”. Deep learning is an ANN having one or more hidden layers.
- Line 173 “Deep learning is ideal for big data…” DL can achieve higher accuracy if trained on big data.
- Line 178. “each neuron in ANN uses a simple classifier model” you mean “activation functions”? “weighted sup to provide a decision tree” – please, clarify your expression.
- Line 181 Cannot agree with these phrases “the class prediction is stored in the second layer”, “the third layer between them are hidden layers”. DNN has input (the first layer), output (the last layer), and hidden layers that may be from second to as many as you want to have.
- Section 3 is strange. DNNs are used in all the tasks: NLP, Chatbots (that are goes under NLP), etc. Please, think about how to restructure your paper.
- Line 251. Can you ground your statement “A chatbot can help fight depression and can be used to engage patients and support them at any time” with a reference? I personally think that similar patients cannot be “given to” chatbots: chatbots are not 100% accurate and their improper answers can lead patients to much more painful consequences. When writing about the advantages of such chatbots do not forget their other side.
- Line 262 “It (chatbot) diagnoses the disease and provides general information…”. See my comment No. 12.
- The topic “ML in Healthcare Communication” made me expect more details about different machine learning methods. The paper is focused on different tools that those tools are used for.
- You try to overview many tools, but present very detailed information (some numbers) only about some of them without any explanation of why this information is more important than the other.
- In Section 5 (about Radiology), Section 6 (Educational systems), Section 7 (Emergency response) you are talking about the chatbots again. Why they are not overviewed in Section 4? You need to rethink the structure of your paper.
- Line 408 “Support Vector Machine (SVM)”. Please, use the full title of the method when you present it for the first time (line 138) and then only abbreviations.
- Since not all the tools are clearly presented, sometimes I do not see the connection between the description and the section. E.g., you are talking about the chatbots under Section 6 (Education System), but I do not see how some of these examples are related to the Educational System.
- The only given schema is about the chatbot. Why such much importance is given to this particular schema (that is not invented by you) in the scope of all machine learning tools in Healthcare Communication?
Author Response
Reviewer 3
We would like to thank the Reviewer for spending time on improving this work. Our answers to the comments are marked in blue fonts while corrections in the revised manuscript are marked in red fonts.
The research paper is an overview of different ML tools used in Healthcare. Despite it contains important information on this topic, some concepts (about the deep learning methods) are not clearly presented. The paper needs to be restructured to be easily understandable.
My comments:
Line 53 “A neural network works similar to how the human brain works”. Well, the artificial neural network is a very very simplified human brain model.
Authors: We rephrased the statement as follows:
“A neural network is a model aspiring to mimic how the human brain works.”
Line 80. Stemming is suitable for EN or agglutinative languages; the other languages need lemmatization.
Authors: We added the related statements as follows:
“Stemming is therefore suitable for agglutinative languages, while other languages need lemmatization.”
Line 83. Linguistic NLP systems (especially word/sentence embeddings-based) usually rely on the assumption that words forming sentences/expressions/texts are conceptual and meaningful.
Authors: The statement is corrected as follows:
“Linguistic NLP systems read words as a symbol combined based on grammatical rules, and usually rely on the assumption that words forming sentences/expression/texts are conceptual and meaningful.”
Line 85. With the era of deep neural networks word n-gram features are replaced with the word or sentence embeddings.
Authors: Corrected
Line 139. CNN does not learn “non-linear complex decision rules”. This neural network approach (as any neural network method) learns only a set of parameters.
Authors: Corrected
“The second model is a CNN that learns a set of parameters based on the raw abstract and title …”
Line 142. Any neural network (CNN, FFNN, RNN, etc.) is a typical supervised ML example that needs the annotated data. Despite if the preparation of the training data is manual or automatic (using rule-based approaches), it does not mean that CNN uses rule-based features. Please, clarify the explanation.
Authors: We agree and phrases regarding “CNN uses rule-based features” are removed.
Line 172 I cannot agree with the statement: “Deep learning is a section of machine learning that is composed of multiple artificial neural networks”. Deep learning is an ANN having one or more hidden layers.
Authors: The statement is corrected as follows:
“Deep learning is an artificial neural networks (ANN) having one or more hidden layers.”
Line 173 “Deep learning is ideal for big data…” DL can achieve higher accuracy if trained on big data.
Authors: Corrected.
Line 178. “each neuron in ANN uses a simple classifier model” you mean “activation functions”? “weighted sup to provide a decision tree” – please, clarify your expression.
Authors: We rephrased the expression as follows:
“Each neuron uses activation functions to output a decision signal according to the weighted sum.”
Line 181 Cannot agree with these phrases “the class prediction is stored in the second layer”, “the third layer between them are hidden layers”. DNN has input (the first layer), output (the last layer), and hidden layers that may be from second to as many as you want to have.
Authors: We rephrased the related statements as follows:
“The first layer is for input, while the last layer is for output, with hidden layers from the second layer to as many layers as required.”
Section 3 is strange. DNNs are used in all the tasks: NLP, Chatbots (that are goes under NLP), etc. Please, think about how to restructure your paper.
Authors: We have divided Section 3 Healthcare communication into four subsections: 3.1 Chatbot, 3.2 Radiology, 3.3 Education system and 3.4 Emergency response.
Line 251. Can you ground your statement “A chatbot can help fight depression and can be used to engage patients and support them at any time” with a reference? I personally think that similar patients cannot be “given to” chatbots: chatbots are not 100% accurate and their improper answers can lead patients to much more painful consequences. When writing about the advantages of such chatbots do not forget their other side.
Authors: We noted the concern of the Reviewer and decided to remove this statement from the revision.
Line 262 “It (chatbot) diagnoses the disease and provides general information…”. See my comment No. 12.
Authors: Related statements are removed.
The topic “ML in Healthcare Communication” made me expect more details about different machine learning methods. The paper is focused on different tools that those tools are used for.
Authors: The original title should be “Application of Artificial Intelligence and Machine Learning in Healthcare Communication”. Since the journal has a word limit of the title (five words), we have to rephrase it as “Machine Learning in Health Communication”. Anyway, Section 2 is about machine learning methods.
You try to overview many tools, but present very detailed information (some numbers) only about some of them without any explanation of why this information is more important than the other.
Authors: In this topical review, we aim at exploring different AI/ML-assisted tools for healthcare communication. Since each tool is clinical or patient-specific, it is difficult to compare them.
In Section 5 (about Radiology), Section 6 (Educational systems), Section 7 (Emergency response) you are talking about the chatbots again. Why they are not overviewed in Section 4? You need to rethink the structure of your paper.
Authors: We have regrouped different sections together for healthcare communication.
Line 408 “Support Vector Machine (SVM)”. Please, use the full title of the method when you present it for the first time (line 138) and then only abbreviations.
Authors: Corrected.
Since not all the tools are clearly presented, sometimes I do not see the connection between the description and the section. E.g., you are talking about the chatbots under Section 6 (Education System), but I do not see how some of these examples are related to the Educational System.
Authors: In the section of Education System, there are some applications using chatbot as a communication tool. This is clearer when we group the sections together.
The only given schema is about the chatbot. Why such much importance is given to this particular schema (that is not invented by you) in the scope of all machine learning tools in Healthcare Communication?
Authors: We agree that there are other schema apps for healthcare communication. However, we mainly focus on chatbot because it is a timely and novel topic with huge potential in patient education/care. It is easy to build incorporating with various AI-assisted features. This is very important for the readers who are medical professionals in healthcare but not familiar with machine learning.
Round 2
Reviewer 1 Report
The authors improved the manuscript.
However, Figure 3 should be improved the resolution quality.
Author Response
We would like to thank the Reviewers again for their insightful comments on improving this work.
Reviewer 1
The authors improved the manuscript. However, Figure 3 should be improved the resolution quality.
Authors: We increased the resolution of Figure 3 in the revised manuscript.
Reviewer 2 Report
Dear author!
I have read and checked your new version. I noticed you guys have better edits compared to the old version.
Author Response
We would like to thank the Reviewers again for their insightful comments on improving this work.
I have read and checked your new version. I noticed you guys have better edits compared to the old version.
Authors: Thank you very much for spending time on reviewing our work.
Reviewer 3 Report
Thank you for addressing some of my previous comments. However, I still think that the paper needs restructuring to be easier readable.
Author Response
We would like to thank the Reviewers again for their insightful comments on improving this work.
Reviewer 3
Thank you for addressing some of my previous comments. However, I still think that the paper needs restructuring to be easier readable.
Authors: We further reorganized and rephrased the names of subsections for a better content flow of the manuscript.
Round 3
Reviewer 3 Report
Thank you for considering my major comments.